# PTSD (Posttraumatic Stress Disorder) in Teachers: A Mini Meta-Analysis during COVID-19

**DOI:** 10.3390/ijerph20031802

**Published:** 2023-01-18

**Authors:** Nahia Idoiaga Mondragon, Idoia Legorburu Fernandez, Naiara Ozamiz-Etxebarria, Beatriz Villagrasa, Javier Santabárbara

**Affiliations:** 1Department of Developmental and Educational Psychology, University of the Basque Country UPV/EHU, 48940 Leioa, Spain; 2Department of Didactics and School Organization, University of the Basque Country UPV/EHU, 48940 Leioa, Spain; 3Psychogeriatry, CASM Benito Menni, 08830 Sant Boi de Llobregat, Spain; 4Centro de Investigación Biomédica en Red de Salud Mental (CIBERSAM), Ministry of Science and Innovation, 28029 Madrid, Spain; 5Department of Microbiology, Pediatrics, Radiology and Public Health, University of Zaragoza, C/Domingo Miral s/n, 50009 Zaragoza, Spain; 6Aragonese Institute of Health Sciences (IIS Aragón), 50009 Zaragoza, Spain

**Keywords:** teachers, COVID-19, post-traumatic stress disorders, prevalence, meta-analysis

## Abstract

Background: Since March 2020, when the World Health Organization (WHO) declared the COVID-19 pandemic, in order to stop the spread of the virus, unprecedented measures were taken worldwide. One of the most important measures was the closure of schools and educational centers around the world in 2020, and very extreme health protocols have been in place in educational centers since they were reopened. From early childhood education to universities, teachers first had to adapt in a short period time to online classes and then continuously readapt to new protocols according to the pandemic situation. This academic environment, in addition to the pandemic situation itself, has favored the emergence of mental disorders such as Post-Traumatic Stress Disorder (PTSD). Materials and Methods: Medline via PubMed and other databases were searched for studies on the prevalence of PTSD in teachers from 1 December 2019 to 1 October 2022. A total of five studies were included in this review. Our results show a prevalence of PTSD of 11% reported by teachers. No subgroups nor meta-regression analyses were performed due to the insufficient number of studies available. Conclusions: The results suggest that teachers are suffering from PTSD, so it is important to carry out more studies worldwide. Similarly, measures to improve the mental health and well-being of teachers during the pandemic and post-pandemic periods are needed.

## 1. Introduction

On 11 March 2020, the World Health Organization (WHO) officially declared the outbreak of COVID-19 disease a pandemic [1]. Since then, to stop the spread of the virus, countries around the world implemented unprecedented public health measures [2,3]. Among these measures, the closure of schools and universities, the change in teaching processes to ensure social and physical distancing, or new security protocols for educational environments were the most repeated actions worldwide [4,5,6]. All of these changes, which took place in a very short time, could have had psychological consequences since the teachers were not prepared to face such a situation.

Firstly, the teachers suffered a significant extension of their working hours [7,8] and continuous changes in protocols in education [9]. In addition, they also had to adapt to new technologies or online classes for which many of these professionals had no prior training [10].

Indeed, when the schools were reopened, they had to face the risk of infection caused by face-to-face teaching [11]. In this context, it cannot be forgotten that teaching is a profession of great human interaction that has been seriously altered [12,13]. In the pandemic era, teachers should have kept their distance from their students, something that makes it difficult to interact and relate with your students, especially with younger children (kindergarten, preschool, elementary school) [14,15]. In addition, they often had to be the support for many children [16,17] and families [18,19] who were being affected by the pandemic on an emotional, social, or economic level, with the extra burden this places on their work as well. In addition to this situation, they had all the demands of a teaching job, so they had to work above their responsibilities and with little or no support.

Similarly, in addition to this stressful situation experienced by teachers in particular, the stress symptomatology brought on by the pandemic itself has to be considered as well [20]. The symptomatology was created by the panic at the fear of illness, the sadness at bereavement, the anguish at isolation, the shock at unemployment, and uncertainty and fear of the future. As a consequence of these and other stressors, meta-analyses of the scientific evidence had already pointed out that teachers were suffering significant levels of depression, anxiety, and stress during this pandemic [21]. In fact, in this stressful situation, both teachers’ physical and mental health is put at risk, leading to the possible onset of post-traumatic stress symptoms. Post-traumatic stress disorder (PTSD) is understood as a state of psychological imbalance following exposure to exceptionally threatening or horrific events and is characterized by a typical symptomatic pattern of intrusions, the persistence of trauma, the avoidance of relevant stimuli, emotional numbing, and physiological hyperarousal [22,23]. This symptomatologic presentation may include fear-based re-experiencing, emotional and behavioral changes, dysphoric moods, and negative effects on cognition [24]. PTSD can generate significant functional impairment, with worsening work performance and family and social relationships. The teachers may have suffered from this symptomatology because they have been on the front lines of this traumatic situation, and it is possible that although the great impact of the pandemic has passed, they still feel fear, and emotional changes, and all this has also influenced them cognitively. Previous studies have shown that participants with high levels of psychological distress often develop PTSD symptoms. In addition, this unforeseen situation may have caused psychological distress among individuals which could result in considerable psychological stress and aggravate PTSD symptoms. The fear of contagion of inadequate magnitude can cause PTSD [25].

During the COVID-19 pandemic, PTSD has been analyzed primarily for symptomatology among health professionals [26], and even among the general population [27,28,29], but no meta-analysis has investigated PTSD in teaching professionals. The COVID-19 pandemic has had an impact on education and the way that teachers engage with their students, in addition to the individual and collective traumatic nature of this global health crisis [30]. Therefore, the current meta-analysis aims to research the evidence on the prevalence of PSTD among teachers. Therefore, the aim of the present meta-analysis is to analyze the prevalence of posttraumatic stress in teachers in different countries during the pandemic. Moreover, this research also aims to know if there are variables that have influenced the incidence of PTSD. This research question follows the FINER (Feasible, Interesting, Novel, Ethical, and Relevant) framework [31].

## 2. Materials and Methods

This study was conducted in accordance with the PRISMA guidelines for reporting systematic reviews and meta-analyses [32] (Appendix A Appendix A).

### 2.1. Search Strategy

According to the Campbell Collaboration [33], two researchers (JS and BV) searched for all cross-sectional studies reporting the prevalence of post-traumatic stress disorder published from 1 December 2019 through 31 December 2021, using MEDLINE via PubMed. The search proceeded as follows:

(covid [tiab] OR covid-19 [tiab] OR coronavirus [tiab] OR SARSCoV-2 [tiab] OR “Coronavirus” [Mesh] OR “severe acute respiratory syndrome coronavirus 2” [Supplementary Concept] OR “COVID-19” [Supplementary Concept] OR “Coronavirus Infections/epidemiology” [Mesh] OR “Coronavirus Infections/prevention and control” [Mesh] OR “Coronavirus Infections/psychology” [Mesh]) AND (“Post-traumatic stress” [Mesh] OR “Posttraumatic stress” [Mesh] OR PTSD [Mesh])

No language restriction was made. References from selected articles were inspected to detect additional potential studies. We then performed a manual search of the “grey literature” (e.g., medRxiv or Google Scholar) to detect other potentially eligible investigations [34]. Any disagreement was resolved by consensus among a third and a fourth researcher (NO-E and NI), according to Harrer et al. [34].

### 2.2. Selection Criteria

Studies were included if: (1) they reported cross-sectional data on the prevalence of post-traumatic stress disorder, or sufficient information to compute this, conducted during the COVID-19 outbreak; (2) focused on teachers; (3) included a validated or reliable instrument to assess PTSD; or (4) the full text was available.

We excluded studies focusing only on community-based samples of the general population or specific samples that were not teachers (e.g., students, medical professionals, patients), as well as review articles.

A pre-designed data extraction form was used to extract the following information: country, sample size, the proportion of women, average age, response rate and sampling methods, and also the instruments used to assess PTSD and their prevalence rates.

### 2.3. Methodological Quality Assessment

To assess the quality of the studies, a risk of bias tool proposed by Loney et al. [35] was used for the systematic review of specific prevalence studies. Quality was evaluated based on eight criteria, each with a score from 0 to 1. In this methodological evaluation system used to qualify the studies, one point was given for each of the following criteria presented: (1) a random sample or complete population was used; (2) there was an unbiased sampling frame (i.e., census data); (3) there was an adequate sample size (>300 subjects); (4) standard measurements were used; (5) the outcomes were measured by unbiased rates; (6) there was an adequate response rate (>70%) and there was a description of losses; (7) confidence intervals and subgroup analysis were reported; and (8) the study subjects were described. The total score could range from 0 (poor quality) to 8 (high quality). Based on the total score, studies were qualified as low risk of bias (6–8), moderate risk (4–5), or high risk of bias (0–3).

Any disagreements that arose between the reviewers were resolved through discussions or by further discussion with a third and fourth researcher (NO-E and NI) [34].

### 2.4. Data Extraction and Statistical Analysis

A generic inverse variance method with a random effects model was utilized [36], using double arcsine transformation of proportion to account for the variability and heterogeneity of prevalence rates among the included studies [37]. The main outcomes were presented in proportion format with a corresponding 95% confidence interval (95%CI) along with statistical heterogeneity results. The Hedges *Q* statistic was reported to check heterogeneity across studies, with the statistical significance set at *p-*value < 0.10. The *I^2^* statistic and 95% confidence interval was also used to quantify heterogeneity [38]. Values between 25–50% are considered as low, 50–75% as moderate, and 75% or more as high [39]. If heterogeneity was present, no subgroups nor meta-regression analyses were performed due to an insufficient number of studies available (*k = 3*) [40,41].

Research methodologists have found that the conventional funnel plots to assess biases in meta-analyses are inaccurate for proportion studies [42]. In the meta-analysis of proportion studies, which was our approach, the fail-safe N value represents the better approach for analytically representing publication bias [43]. This statistic is recommended when there are fewer than 10 studies in the meta-analysis [41,44], and indicates the number of non-significant, unpublished (or missing) studies that would need to be added in the meta-analysis to reduce an overall statistically significant result to non-significance. There is confidence in the summary conclusions if this number is large relative to the number of observed studies [43].

All statistical analyses will be conducted by one study (JS) and will run with R [45] with the *metaprop*, *metafor*, and *dmetar* packages for meta-analysis, and *p*-values will be reported as two-sided, with 0.05 accepted as statistically significant except where otherwise indicated.

## 3. Results

Figure 1 shows the flowchart of the search strategy and study selection process. A total of 723 records were initially identified from Medline via PubMed, and six extra records were then added after a manual search in another source (Google Scholar). A total of 719 records were excluded after the first screening of the titles and abstracts. After reading the remaining 10 articles in full, we finally included five in our meta-analysis [11,25,46,47,48]. Exclusion reasons are detailed in Figure 1.

Table 1 gives a descriptive overview of the global characteristics of the included studies. The majority of studies were carried out in China; two studies were focused on university teachers [11,48], two on school teachers [25,47], and one on all teachers [46]. The sample size ranged from 67 to 818,529 participants, and only one reported an age mean [11]. All studies included both men and women, and the percentage of women ranged from 46.3% to 78.9%. All studies used standardized and/or validated scales. Outcome assessments of the included studies showed that in the study of Domuschieva-Rogleba & Savcheva (2020), there were 11 cases (16.4%) of prevalence; in the study of Fan et al. (2020) measured with IES-R and criteria ≥ 1.5, there were 405 cases (24.5%); in the study of Kukreti et al. (2021) measured with DSM-5 (PCL-5) and criteria there were ≥31,321 cases; (12.3%), in the study of Lizhi et al. (2021) measured with PC-PTSD and criteria there were ≥4, 93 cases (0.5%); and Liang et al. (2022), who measured with DSM-5 (PCL-5) and criteria, there were ≥33, 978 cases (8.9%). They were conducted by using online questionnaires, and, of those reporting sampling methodologies, only one used non-randomized methods and reported the response rate [11].

Regarding the quality of the studies (Table 2), one of them was classified as having a low risk of bias [11], while two studies were qualified as having a high risk of bias [47,48]. The main limitation present in all studies was that the absence of PTSD measured by unbiased rates could not be guaranteed, due to the use of online surveys.

Only five studies reported the prevalence of PTSD data. The estimated overall prevalence of PTSD was 11% in teachers (95% CI: 3–22%), with significant heterogeneity between studies (*Q* test: *p-*value < 0.001; *I*^2^ = 99.8% [47,48] 95% CI: 99.8–99.9% [47,48]) (Figure 2). No subgroups nor meta-regression analyses were performed due to an insufficient number of studies available (*k =* 5).

Excluding each study one by one from the analysis did not substantially change the pooled prevalence of PTSD, which varied from between 8% (95% CI: 1–19%), with Fan et al. [11] excluded, and 15% (95% CI: 8–23%), with Lizhi et al. [46] excluded (Figure 3). This indicates that no single study had a disproportional impact on the overall PTSD prevalence.

A publication bias was indicated by a fail-safe N equal to 427, suggesting that 427 studies with the null result are necessary to reduce the observed overall prevalence to non-significance. This would indicate the absence of publication bias.

## 4. Discussion

The COVID-19 pandemic had an unprecedented and traumatic impact on societies all around the world, worsening mental health in general. Some reviews highlight that people affected by this pandemic situation have a high epidemiological burden of several mental health disorders such as depression, anxiety disorders, sleep disorders, or PTSD that may cause a significant deterioration in quality of life [49,50]. It is therefore legitimate and necessary to analyze to what extent the pandemic era is creating PTSD in citizens in general, and also in important groups such as teachers. In this context, the present research provides an up-to-date meta-analysis of studies reporting the prevalence of PTSD in teachers during the COVID-19 pandemic. Our meta-analysis is based on five studies, and to the best of our knowledge, we are the first to report overall prevalence rates of PTSD in teachers.

The findings of this meta-analysis show that teachers report a pooled prevalence of PTSD of 11%. Some previous systematic reviews and meta-analyses have been conducted to report the prevalence of PTSD in the general population and other professional collectives in the COVID-19 era. In the general population, Cenat el at. [27], in a meta-analysis of 13 papers, found a prevalence of PTSD of 21.94%. Salehi et al. [28], analyzing 35 papers, found a PTSD symptomatology of 18%. Qiu et al. [29] identified 76 articles and found a prevalence of 28.34%. Finally, with a more general perspective, Yuan et al., [51] analyzed PTSD after infectious disease pandemics of the twenty-first century, including COVID-19, but also other diseases such as sudden acute respiratory syndrome (SARS), H1N1, Poliomyelitis, Ebola, Zika, Nipah, Middle Eastern respiratory syndrome coronavirus (MERS-CoV) and H5N1. These authors analyzed 77 papers and found a prevalence of 22.6%. Therefore, it seems evident that the prevalence of PTSD in a pandemic or post-pandemic situation is lower in teachers than in the general population.

Previous research has also analyzed PTSD in different professional or socio-demographic groups. Cénat et al. [27] found a prevalence of 22.43% for citizens and 20.91% for healthcare workers. Salehi et al. [28] found a prevalence of 18% among healthcare workers, 29% among survivors, and 12% among the general population. Ghahramani et al., in 2022 [52] found a prevalence of 37% in a different type of healthcare worker. Finally, Qiu et al. [29] found a prevalence of PTSD of 36.30% in COVID-19 patients, 29.22% among healthcare workers, 24.47% in suspected cases of COVID-19, 27.13% in the general population, and 29.39% in a teacher/student collective. However, in their analysis, students and teachers were taken as a single group, as they found only one study that analyzed the prevalence of teachers. Yuan et al. [51] also found that healthcare workers had the highest prevalence of PTSD (26.9%), followed by infected cases (23.8%) and the general public (19.3%). Therefore, in this case, the prevalence of 11% of teachers seems to be lower than that of other professional groups such as healthcare workers and that of the general population (although in many studies it is very close), and even that of the teacher/student group.

Nevertheless, despite an overall prevalence of 11%, some of the included studies in this review reported a high prevalence of PTSD, up to 24.55% among teachers [11]. Moreover, although in our study there were not enough samples to make comparisons by subgroups, it is true that almost all of the studies found were from China (excluding one) and, in some of the other systematic reviews, a higher prevalence of PTSD in Western Pacific region countries was also found [28,29]. Furthermore, pre-pandemic research estimated that between 6.4% and 6.8% of the global population would show symptoms of PTSD in their lifetime, levels that have been exceeded, including in the teaching profession [53,54,55]. Therefore, it is essential to take this increase into consideration because it has been proven that PTSD is an anxiety disorder that can significantly affect people’s quality of life [28], as it is associated with an increased risk of future physical and psychiatric comorbidities such as depression, substance use, suicide and chronic disease [56,57]

Similarly, it should also be noted that these levels of PTSD may take longer to emerge, as those serving traumatized populations, such as therapists, social workers, and teachers are more vulnerable to “shared trauma” and “compassion fatigue” [49,58,59]. In other words, teachers might be indirectly experiencing the trauma of the populations they serve, resulting in emotional, physical, and cognitive responses that have not yet surfaced [50].

Therefore, from these results, we would also like to propose several implications for practice. First, teachers must be aware of the psychological consequences of the COVID-19 pandemic, including PTSD. They can be prepared with effective strategies on how to handle the commonly expected PTSD during the ongoing COVID-19 pandemic or similar disasters in the future, working with previously effective coping mechanisms. That is, further mental health strategies aiming to improve the long-term mental health outcomes of teachers should be integrated into their training programmes. Similarly, the education authorities in each country should also regularise routine testing to ensure the mental health of education professionals, thus detecting deterioration in specific crisis situations.

The greatest strength of the present study is that, to our knowledge, no meta-analysis has been carried out that focuses on teachers’ symptomatology of PTSD during the COVID-19 pandemic. This is why this study may provide the basis for further studies along this research line. In this respect, it would be interesting to monitor the levels of PTSD among teachers during the post-pandemic period and to analyse its trajectory. It would also be interesting to carry out studies on PTSD among different types of teachers, from early childhood educators to university lecturers. It would also be interesting to triangulate these results with the levels of PTSD that students (whether children, adolescents, or young university students) may be suffering from with the levels of PTSD of their teachers. Moreover, another strength of this research is that it was determined that there is no publication bias in the estimation of the pooled prevalence of PTSD.

However, some limitations should be considered when interpreting our results. The main limitation of this study is the quantity of the available literature, as only five relevant papers were found, and this may affect the power of the test used. However, some research has shown that meta-analyses of a few studies will still be able to provide important information [60], and this does not preclude the validity of meta-analysis [61]. In this sense, we used the *I^2^* confidence interval for the assessment of heterogeneity instead of only the *Q* contrast, which is recommended in meta-analyses of a few individual studies [62]. Similarly, with regard to the quality of the analyzed papers, although all papers present only a moderate risk of bias, all studies reported the absence of a random sample. In addition, it could not be guaranteed that the results were measured by unbiased rates, since online surveys were used. Furthermore, although most of them use validated instruments in one of the cases, this is not the case, however, the results prove that this study is not influential, as demonstrated in the influence analysis. The different validated instruments included a screening tool such as a PC-PTSD which could overestimate the prevalence of PTSD, but the influence analysis was not affected by that. Finally, since the systematic review requires previous existing scientific publications when evaluating any condition during the pandemic using this methodology, there will be a scarce availability of information and a high risk of including the literature of moderate-to-low methodological quality. Thus, more high-quality studies, specifically using a diagnostic validated instruments, are needed.

## 5. Conclusions

Although we are aware of the limited sample of this meta-analysis, we believe that it has provided a feasible, interesting, novel, ethical and relevant investigation [31], and that it demonstrates that there is a need for more research on PTSD and teachers worldwide, since a significant proportion of teachers may be suffering from PTSD. It is therefore urgent to investigate it further, but also to provide teachers with the necessary resources to cope emotionally with this pandemic. Improving the emotional state of teachers in the pandemic and post-pandemic eras would have a direct impact on society, as it would directly influence the quality of education and the emotional state of students and future generations. We must remember that the social role of teachers is paramount in society, and even more so when faced with traumatic social challenges such as the COVID-19 pandemic.

## Figures and Tables

**Figure 1 ijerph-20-01802-f001:**
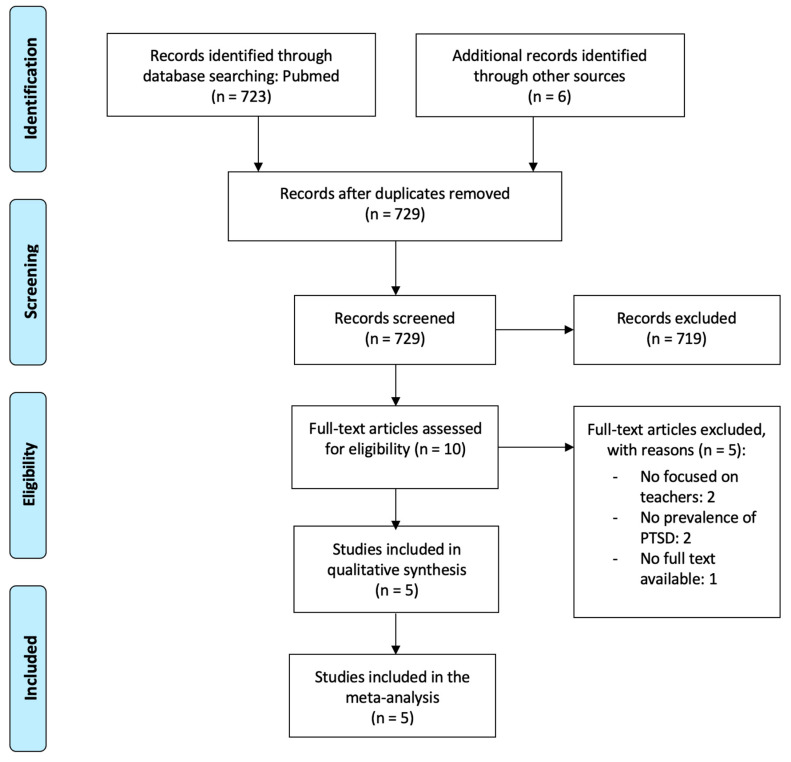
Flowchart of the study search and selection process.

**Figure 2 ijerph-20-01802-f002:**
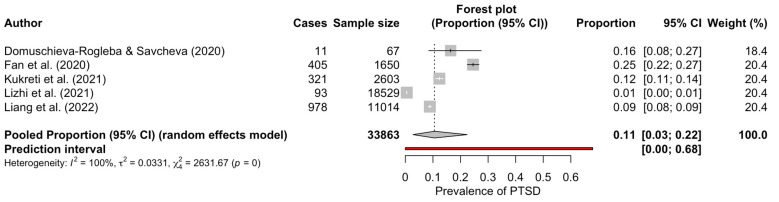
Forest plot for the prevalence of PTSD among teachers [11,25,46,47,48].

**Figure 3 ijerph-20-01802-f003:**
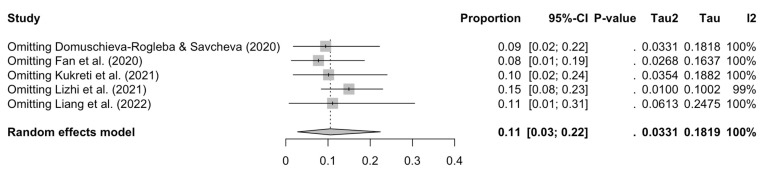
Sensitivity forest plot for the prevalence of PTSD among teachers [11,25,46,47,48].

**Table 1 ijerph-20-01802-t001:** Characteristics of the studies included in the meta-analysis.

First Author (Publication Year)	Sample Country	Population	Sample Size (n)	Mean Age (SD)	Females (%)	Response Rate (%)	Sampling Method	Quality Assessment
Domuschieva-Rogleba & Savcheva (2020) [48]	Bulgaria	University teachers	67	NR	46.3%	NR	NR	2
Fan et al. (2020) [11]	China	University teachers	1650	40.3 (8.3)	51.8%	80%	Random	6
Kukreti et al. (2021) [25]	China	School teachers	2603	NR	71.6%	NR	Non-probabilistic	5
Lizhi et al. (2021) [46]	China	All teachers	18,529	NR	78.9%	NR	NR	5
Liang et al. (2022) [47]	China	Primary and middle school teachers	11,014	NR	71.3%	NR	NR	3

Abbreviations: SD, standard deviation; NR, not reported.

**Table 2 ijerph-20-01802-t002:** Quality assessment.

Study	1	2	3	4	5	6	7	8	Total
Domuschieva-Rogleba & Savcheva (2020) [48]	0	0	0	0	0	0	1	1	2
Fan et al. (2020) [11]	1	0	1	1	0	1	1	1	6
Kukreti et al. (2021) [25]	0	0	1	1	0	1	1	1	5
Lizhi et al. (2021) [46]	0	1	1	1	0	0	1	1	5
Liang et al. (2022) [47]	0	0	1	1	0	0	0	1	3

Abbreviations: (1) Random sample or entire population; (2) Unbiased sampling frame (census data); (3) Adequate sample size (>300 subjects); (4) Standard measures were used; (5) Outcome measured by unbiased raters; (6) Adequate response rate (>70%) and description of losses; (7) Confidence intervals and subgroup analysis; (8) Study subjects described.

## Data Availability

Not applicable.

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
