# Peer review of "PTSD (Posttraumatic Stress Disorder) in Teachers: A Mini Meta-Analysis during COVID-19"

_ijerph, 2023, doi:10.3390/ijerph20031802_

Round 1

Reviewer 1 Report (New Reviewer)

It is not clear why 719 studies were excluded from the meta-analysis. However, justifications were given even for 5 studies excluded at other stages. Only 5 of the 729 publications were included in the meta-analysis. This is not a sufficient number.

The shortcoming requiring major revision is that the meta-analysis was conducted with only 5 studies. When the meta-analysis is expanded with new studies to be added, the discussion and conclusion sections of the study should also be expanded. Especially in the discussion section, details can be given within the scope of the findings of the new studies to be added.

Author Response

Dear Editor and Reviewers,

We would like to thank you for considering our manuscript for publication in Environmental and public health.  We have found your comments and insights to be most constructive and helpful and we have tried to address the issues you proposed.

In addition, we attach a document setting out comment by comment the changes that have been made.

Reviewer 2 Report (New Reviewer)

A number of successful aspects of the article can be highlighted.

It is used The Preferred Reporting Items for Systematic reviews and Meta-Analyses (PRISMA).

The discussions are developed in accordance with current research.

Some aspects that require improvements can also be noted.

The objectives nad questions of the research are not very clearly defined.

The theoretical part is briefly elaborated. 

Very few studies are analyzed, only 5.

Author Response

Dear reviewer

We would like to thank you for considering our manuscript for publication in Environmental and public health.  We have found your comments and insights to be most constructive and helpful and we have tried to address the issues you proposed.

We attach to this letter the word document with change “change editor” tool, where all the changes that we have made are. Moreover, in the following text we will expose comment by comment  the changes that have been made:

The discussions are developed in accordance with current research.

 Thank you very much for these comments.

Some aspects that require improvements can also be noted.

The objectives and questions of the research are not very clearly defined.

Thank you. We have clarified the objectives and the research question.

The theoretical part is briefly elaborated. 

Thank you very much, we have extended the theoretical information.

Very few studies are analyzed, only 5.

Thank you. We assume as an important limitation of the manuscript that only 5 studies have been selected for the meta-analysis and we extend the discussion by adding this limitation but also justifying (with new bibliographic references) that meta-analyses with few studies do not necessarily provide less scientific evidence than those that include a larger number of selected articles.

The new discussion paragraph will be the following (the new modifications are highlighted in green):

However, some limitations should be considered when interpreting our results. The main limitation of this study is the quantity of the available literature, in fact, only five relevant papers were found, and this may affect the power of the test used. However, some research has shown that meta-analyses of a few studies will still be able to provide important information (Goh, 2016), and this does not preclude validity of meta-analysis (Valentine & Pigott, 2010). In this sense, we used the I2 confidence interval for the assessment of heterogeneity instead of only the Q contrast, which is recommended in meta-analyses of few individual studies (Thorlund et al., 2012). Likewise with regards to the quality of the analyzed papers, although all papers present only a moderate risk of bias, all studies reported the absence of a random sample…

References:

Goh JX, Hall JA, Rosenthal R. Mini meta-analysis of your own studies: some arguments on why and a primer on how. Soc Personal Psychol Compass. 2016;10:535-49.

Thorlund K, Imberger G, Johnston BC, Walsh M, Awad T, Thabane L, et al. Evolution of heterogeneity (I2) estimates and their 95% confidence intervals in large meta-analyses. PLoS One. 2012;7:e39471.

Valentine, J.C.; Pigott, T.D.; Rothstein, H.R. How many studies do you need? A primer on statistical power for Meta-analysis. JEBS 2010, 35, 375.

Reviewer 3 Report (New Reviewer)

The manuscript is focused on the problematic of covid-19 syndrome among teachers. Authors used analysis of literary sources. The text is written in understandable form and it includes minimum typographical errors. The text is written on high level and I have got some comments of minor character.

1. I have got positive comment toward theoretical part of the manuscript, which is  brief and provides actual and needed kinds of information for the whole understanding of the manuscript.

2. Crucial comment is regarding to methodological part of the manuscript. Authors wrote, that the choosing of studies in the analyses was without any limit.  I do not understand this decision, because I do not think, that some local journals provide so quality kinds of information in comparison with journals included in some databases line Web of Science or SCOPUS. Please explain your decision.

3. Please read carefully, mainly references, and revise them according guidelines for authors.

I hope my comments are helpful.

Author Response

Dear reviewer

We would like to thank you for considering our manuscript for publication in Environmental and public health.  We have found your comments and insights to be most constructive and helpful and we have tried to address the issues you proposed.

We attach to this letter the word document with change “change editor” tool, where all the changes that we have made are. 

We hope that with all the changes we have made, the article has improved significantly.
Thank you

Reviewer 4 Report (New Reviewer)

The article entitled “PTSD (Posttraumatic Stress Disorder) in Teachers. A Meta-Analysis during COVID-19” illustrates an interesting and potentially valuable contribution to the literature on psychological disorders. It adds to the latter the perspective of a population that is most likely to be understudied. In my modest opinion, a few changes may need to be made before publication is warranted though.

 The title does not reflect the content of the study. The authors did not perform a meta-analysis. Theirs is mostly a literature review.

 Abstract

The following statement may be rephrased to represent the researchers’ main hypothesis:  “This academic environment, added to the pandemic situation itself, has created PTSD (post-traumatic stress disorders) in teachers.” In its current format, it is awkward.  The authors have yet to address the results of their study.

The following statement is a bit of a stretch since only 11% of the estimated population displayed evidence of PTSD.

 Introduction

The introductory section leaves the reader asking a few questions. For instance, how does the definition of PTSD apply to teachers? Is there a different way to classify the distress that teachers experienced? When does distress become PTSD?  The review of the literature is rather modest and unsatisfactory.

Methodology and Results

The results of the literature review performed by the authors uncovered only 5 studies. This very small number of studies may be an indication that the attention of the scientific community was devoted to other populations (e.g., nurses and medical doctors). However, it may also indicate that the diagnostic label may not fit well the selected population. Is there an alternative to the selected diagnostic category? Can a larger number of studies be found if the authors look for individual symptoms, such as “distress” or even “severe distress” instead of PTSD?

The description of the procedure used to analyze the data of the 5 studies requires clarification. The authors state that “[o]nly three studies reported prevalence of PTSD data. The estimated overall prevalence of PTSD was 11% in teachers (95% CI: 3–22%), with significant heterogeneity between studies (Q test: p-value < 0.001; I2 = 99.8%) (Figure 2). What does the heterogeneity of the selected studies’ results say about PTSD among teachers? Can any reasonable conclusions be made with such a small sample?

Discussion

The discussion needs to be better connected with the extant literature on mental disorders. Most importantly, a critical examination of the controversy regarding the overuse of clinical diagnostic labels (e.g., PTSD) for a diverse array of individuals should be considered.  Other concerns may also be explored in more depth. For instance, how does the prevalence of PTSD in different populations of professionals relate to the 11% reported here?  What are the implications and remedies of the current findings? 

Author Response

Dear Editor and Reviewers,

We would like to thank you for considering our manuscript for publication in Environmental and public health.  We have found your comments and insights to be most constructive and helpful and we have tried to address the issues you proposed.

We attach to this letter the word document with change “change editor” tool, where all the changes that we have made are. Moreover, in the following text we will expose comment by comment  the changes that have been made:

- The title does not reflect the content of the study. The authors did not perform a meta-analysis. Theirs is mostly a literature review.

We have changed it to: PTSD (posttraumatic stress disorder) in teachers. A revision during COVID-19. Do you think it's better?

Thank you

 Abstract

The following statement may be rephrased to represent the researchers’ main hypothesis:  “This academic environment, added to the pandemic situation itself, has created PTSD (post-traumatic stress disorders) in teachers.” In its current format, it is awkward.  The authors have yet to address the results of their study.

The following statement is a bit of a stretch since only 11% of the estimated population displayed evidence of PTSD.

Thank you, we have changed the sentence

 Introduction

The introductory section leaves the reader asking a few questions. For instance, how does the definition of PTSD apply to teachers? Is there a different way to classify the distress that teachers experienced? When does distress become PTSD?  The review of the literature is rather modest and unsatisfactory.

Thank you, we have expanded the introduction following your suggestions.

Methodology and Results

The results of the literature review performed by the authors uncovered only 5 studies. This very small number of studies may be an indication that the attention of the scientific community was devoted to other populations (e.g., nurses and medical doctors). However, it may also indicate that the diagnostic label may not fit well the selected population. Is there an alternative to the selected diagnostic category? Can a larger number of studies be found if the authors look for individual symptoms, such as “distress” or even “severe distress” instead of PTSD?

The description of the procedure used to analyze the data of the 5 studies requires clarification. The authors state that “[o]nly three studies reported prevalence of PTSD data. The estimated overall prevalence of PTSD was 11% in teachers (95% CI: 3–22%), with significant heterogeneity between studies (Q test: p-value < 0.001; I2 = 99.8%) (Figure 2). What does the heterogeneity of the selected studies’ results say about PTSD among teachers?

Thank you. Despite analyzing few studies, we are able to detect that there is a "statistical" heterogeneity among the different studies analyzed. In fact, it is striking that the only two studies focused exclusively on university teachers (Domuschieva et al.) (Fan et al) are also those that show a higher prevalence of PTSD. We add this finding to the discussion. Can any reasonable conclusions be made with such a small sample?

Can any reasonable conclusions be made with such a small sample?

Thank you. We assume as an important limitation of the manuscript that only 5 studies have been selected for the meta-analysis and we extend the discussion by adding this limitation but also justifying (with new bibliographic references) that meta-analyses with few studies do not necessarily provide less scientific evidence than those that include a larger number of selected articles.

The new discussion paragraph will be the following (the new modifications are highlighted in green):

However, some limitations should be considered when interpreting our results. The main limitation of this study is the quantity of the available literature, in fact, only five relevant papers were found, and this may affect the power of the test used. However, some research has shown that meta-analyses of a few studies will still be able to provide important information (Goh, 2016), and this does not preclude validity of meta-analysis (Valentine & Pigott, 2010). In this sense, we used the I2 confidence interval for the assessment of heterogeneity instead of only the Q contrast, which is recommended in meta-analyses of few individual studies (Thorlund et al., 2012). Likewise with regards to the quality of the analyzed papers, although all papers present only a moderate risk of bias, all studies reported the absence of a random sample…

References:

Goh JX, Hall JA, Rosenthal R. Mini meta-analysis of your own studies: some arguments on why and a primer on how. Soc Personal Psychol Compass. 2016;10:535-49.

Thorlund K, Imberger G, Johnston BC, Walsh M, Awad T, Thabane L, et al. Evolution of heterogeneity (I2) estimates and their 95% confidence intervals in large meta-analyses. PLoS One. 2012;7:e39471.

Valentine, J.C.; Pigott, T.D.; Rothstein, H.R. How many studies do you need? A primer on statistical power for Meta-analysis. JEBS 2010, 35, 375.

Discussion

The discussion needs to be better connected with the extant literature on mental disorders. Most importantly, a critical examination of the controversy regarding the overuse of clinical diagnostic labels (e.g., PTSD) for a diverse array of individuals should be considered.  Other concerns may also be explored in more depth. For instance, how does the prevalence of PTSD in different populations of professionals relate to the 11% reported here?  What are the implications and remedies of the current findings? 

We have tried to better connect the discussion with previous literature. In particular, we have explored both the significance of the prevalence found. On the controversy of the overuse of diagnostic labels in particular in the case of PTSD, we echo this possibility by emphasizing that one of the studies uses a screening tool rather than a diagnostic tool. We also emphasize that the diagnosis of PTSD is associated with a significant impairment in functionality in various areas (work, family, social...), which would not occur with other entities that do not constitute a disorder.

Round 2

Reviewer 1 Report (New Reviewer)

Number of analysed sources are limited with only 5 studies. This is the most important and main limitation of this study.

Conclusion of the papar can be extended with the practical implications of the research findings. And it will be better to give suggestions for further research on the topic.

Author Response

Dear Editor and Reviewers,

We would like to thank you for considering our manuscript for publication in Environmental and public health.  We have found your comments and insights to be most constructive and helpful and we have tried to address the issues you proposed.

We attach to this letter the word document with change “change editor” tool, where all the changes that we have made are. Moreover, in the following text we will expose comment by comment  the changes that have been made:

Number of analysed sources are limited with only 5 studies. This is the most important and main limitation of this study.

Yes you are right, we are aware of this and have included it both in the reviews section and in the final conclusions.

For this reason we have also changed the title of the paper to the following:

PTSD (posttraumatic stress disorder) in teachers. A mini meta-analysis during COVID-19.

This was also done in the following article:

Belvederi Murri M, Respino M, Masotti M, Innamorati M, Mondelli V, Pariante C, Amore M. Vitamin D and psychosis: mini meta-analysis. Schizophr Res. 2013 Oct;150(1):235-9. doi: 10.1016/j.schres.2013.07.017. Epub 2013 Jul 29. PMID: 23906618

Conclusion of the papar can be extended with the practical implications of the research findings. And it will be better to give suggestions for further research on the topic.

Implications for practice and future lines of research are included in the conclusions section.

(x) English language and style are fine/minor spell check required

The English language and the style of the whole document have been checked again to fix the mistakes.

Reviewer 4 Report (New Reviewer)

The authors of the study entitled “PTSD (posttraumatic stress disorder) in teachers. A meta-analysis during COVID-19” have made a series of significant changes to the original manuscript.  In my modest opinion, the changes made are adequate and likely to enhance the article's readership. The authors have been sensitive to the advice given by each of the reviewers. Clarity has improved readers’ ability to understand the content of the revised manuscript (including its rationale and methodology). Thus, I would recommend publication. Of course, additional proofreading may remove some of the remaining grammatical anomalies and improve a few uncommon word choices.  

Author Response

Dear  Reviewer,

We would like to thank you for considering our manuscript for publication in Environmental and public health.  We have found your comments and insights to be most constructive and helpful and we have tried to address the issues you proposed.

We attach to this letter the word document with change “change editor” tool, where all the changes that we have made are. Moreover, in the following text we will expose comment by comment  the changes that have been made:

The authors of the study entitled “PTSD (posttraumatic stress disorder) in teachers. A meta-analysis during COVID-19” have made a series of significant changes to the original manuscript.  In my modest opinion, the changes made are adequate and likely to enhance the article's readership. The authors have been sensitive to the advice given by each of the reviewers. Clarity has improved readers’ ability to understand the content of the revised manuscript (including its rationale and methodology). Thus, I would recommend publication.

Thank you very much

Of course, additional proofreading may remove some of the remaining grammatical anomalies and improve a few uncommon word choices. 

The English language and the style of the whole document have been checked again to fix the mistakes

This manuscript is a resubmission of an earlier submission. The following is a list of the peer review reports and author responses from that submission.

Round 1

Reviewer 1 Report

  1. Meta-analysis of three studies, what is the theoretical and practical value of this study?
  2. How does this study's understanding of meta-analysis differ from that of international academia?

     3.if readers or researchers want to know the PTSD information of teachers during the COVID-19 pandemic, what is the difference between reading this study and consulting the other three studies?

  4.The format of the references needs to be adjusted according to the requirements of the journal.

   5.In “4. Discussion” “However, some limitations should be considered when interpreting our results. The main limitation of this study is the quantity of the available literature, in fact, only three relevant papers were found. Likewise, with regards to the quality of the analyzed papers, athough all papers present only a moderate risk of bias, all studies reported the absence of a random sample.”

In this study, a total of three studies are analyzed, and the analyzed studies also reveal the defects of information collection. What is the fundamental purpose of this study? What is the most valuable thing the study offers readers and researchers?

Reviewer 2 Report

GENERAL

The scope is limited to the prevalence of PTSD alone and the very small number of articles limit the application of the findings.

ABSTRACT

Please write the “Result” section.

Shorten the Background. Add on the Method and Result section.

INTRODUCTION

The impact of PTSD is a bit vague since the pandemic is an ongoing situation.

METHOD

Use the updated version of PRISMA.

The primary search using MedLine alone is less convincing.

There are more tools for quality assessment such as the New Castel Ottawa Scale which is widely used for observational studies.

DISCUSSION

Please focus on the findings of this review rather than comparing with individual studies of different populations.

Please add the strength of this review with regards to the methods used rather than no meta-analyses has been carried out.

Reviewer 3 Report

The COVID-19 pandemic and the global health crisis affect education and how teachers interact with their students. This study was to investigate clear evidence on the prevalence of PTSD among teachers using a meta-analysis approach. The social role of teachers is essential. And in a society faced with a traumatic social course, such as in the context of the COVID-19 pandemic, the part of teachers is even more critical, making the findings of this study very relevant.

In this study, three articles were included in the meta-analysis; it cannot be denied that three is a small number for a typical meta-analysis analysis. However, we believe that the analysis and statistical analysis methods are accurate.

Major Comments

The meta-analysis results showed that the prevalence of PTSD in teachers as a profession was about 10%. In the discussion part, the authors present the results of the meta-analysis analysis for different professions and socio-demographic groups. Compared to these results, the prevalence of PTSD among teachers is lower. The authors discuss the common reasons for this.

The purpose of this study is the prevalence of PTSD in teachers as a profession. Thus, the perspective is placed on the profession. In the discussion part, several examples of the prevalence of PTSD from meta-analytic analyses of other occupational workers are given. If possible, the authors should present more occupation-specific analyses to focus on the prevalence of PTSD in the teaching profession.

From a regional perspective, it is noted that all three of the articles included in the meta-analysis in this study were studies conducted in China. The authors have included some discussion of this regionality in the papers. It would be essential to deepen the consideration of regional differences by comparing the results with those of other regions based on previous studies.

Age and gender are essential factors in studies of PTSD, and among the three papers in this study, one paper reported on age, and three reported on gender (From table1 table2). If possible, a discussion of age and sex differences should be conducted and presented in the article.

Reviewer 4 Report

Overall, the manuscript is well written. However, I am not sure there is any added value of this study. It is not clear to me why the authors were interested in this topic. In the Introduction section, the authors are citing a few studies on the challenges the teachers faced during the pandemic, then  they provided the definition of what PTSD is followed by stating the aim of the paper. Although this logic is not incorrect, I think the manuscript needs to focus on consequences and implications of PTSD for teachers and the students and build their arguments around those topics in order to strengthen the Introduction and increase the interest of the manuscript. One can guess why such a topic might be of interest for the readership but the authors need to build it and include it in the manuscript. For example, the authors might want to add the implications for students and for teachers themselves on a longer term.

Typically at least three databases are used in a systematic-review or meta-analysis and the authors conducted searches only in one database and searched for grey literature in Google Scholar. Also from the keywords it is not clear why the authors focused on covid-19 terms and they did not include PTSD and/or keywords related to teaching. It seems there might have been a different objective and since it did not work, the authors shaped their manuscript into something more in consonance with what they found. If that is the case, the authors should describe the evolution of their research question and and explain how their research question became something different. Research is a reiterative process and sometimes research questions become something different after the searches were conducted and the screening process started.

All those concerns have also repercussions with regards to the Discussion. While I am reading the Discussion I think a different journal could be a better audience for this manuscript since it is very specific to teachers and it does not describe nor discuss public health implications. It seems the methodology is a strength of the manuscript.